# Spatial Risk Distribution of Lumpy Skin Disease in Thailand Based on Maximum-Entropy Modeling

**DOI:** 10.3390/ani15162456

**Published:** 2025-08-21

**Authors:** Kusnul Yuli Maulana, Supitchaya Siriyakhun, Kannika Na-Lampang, Kannikar Intawong, Kenny Oriel A. Olana, Wengui Li, Maytawee Tamprateep, Veerasak Punyapornwithaya

**Affiliations:** 1Faculty of Veterinary Medicine, Chiang Mai University, Chiang Mai 50100, Thailand; kusnulyulimaulana_k@cmu.ac.th (K.Y.M.); kannika.nalampang@cmu.ac.th (K.N.-L.); 2Research Center for Veterinary Biosciences and Veterinary Public Health, Faculty of Veterinary Medicine, Chiang Mai University, Chiang Mai 50100, Thailand; siriyakhun.s@gmail.com; 3Faculty of Public Health, Chiang Mai University, Chiang Mai 50200, Thailand; kannikar.i@cmu.ac.th; 4Faculty of Veterinary Medicine, Visayas State University, Baybay City 6521-A, Philippines; kennyoriel_o@cmu.ac.th; 5The Joint International R&D Center of Veterinary Public Health, College of Veterinary Medicine, Yunnan Agricultural University, Kunming 650201, China; liwengui@ynau.edu.cn; 6Department of Livestock Development, Bangkok 10400, Thailand; maytawee.dld@gmail.com; 7Veterinary Public Health and Food Safety Centre for Asia Pacific, Chiang Mai University, Chiang Mai 50100, Thailand

**Keywords:** lumpy skin disease, ecological niche modeling, risk map, suitability, animal health, Thailand

## Abstract

Controlling lumpy skin disease (LSD) in livestock requires more than understanding its biology, it demands insight into where and why outbreaks occur. This study applied the maximum-entropy-modeling approach to LSD outbreak data from 2021 to 2023 in Thailand, integrating environmental variables such as land cover, normalized difference vegetation index (NDVI), cattle density, and climate factors. Land cover, NDVI, and cattle density emerged as the most influential predictors of disease suitability. The model has an area under the curve value of 0.699 (~0.70) indicating moderate predictive ability. It also has reliable predictive power. Based on this, the model’s performance supports its reliability in identifying environmentally suitable areas for LSD. Central and northeastern region of Thailand were defined as a high-suitability area of LSD outbreak.

## 1. Introduction

Lumpy skin disease (LSD) is a significant transboundary viral disease that primarily affects cattle and buffaloes, with profound implications for animal health, productivity, and the livelihoods of communities that depend on livestock [1,2]. Caused by the lumpy skin disease virus (LSDV), the disease is characterized by fever, skin nodules, emaciation, reproductive disorders, and, in severe cases, death [3,4,5,6], with the severity and presentation of clinical signs varying according to the LSDV strain [7]. LSD imposes considerable economic burdens through production losses, trade restrictions, and the expenses associated with outbreak control and vaccination initiatives [8,9]. In recent years, the disease has increasingly spread across borders, posing a growing threat to the livestock sector in Southeast Asia [10,11,12,13,14]. The ongoing outbreaks in the region have raised concerns regarding the effectiveness of existing prevention and response strategies, particularly considering the complex interactions among livestock movement, vector ecology, and environmental factors.

LSD has emerged as a significant transboundary disease affecting cattle populations in Thailand since its initial detection in 2021, with animal movement from previously affected countries hypothesized as the primary source of introduction. The disease rapidly expanded into central regions and, within a few months, spread across most parts of the country. This pattern of spread was consistent with known risk factors, including smallholder farming systems and limited vector control measures [14,15]. In response, several studies have been conducted to characterize the epidemiological features of LSD outbreaks. These studies primarily focus on temporal trends, outbreak dynamics, and spatial distribution patterns through descriptive analyses and mapping techniques [15,16,17,18]. Furthermore, in recent years, epidemiological techniques based on spatiotemporal prediction and modeling have become essential tools for understanding disease outbreaks and guiding targeted interventions [19,20]. However, although previous research has examined the spatiotemporal dynamics of LSD in Thailand using nationwide surveillance data [15], these analyses have not extended to evaluating the environmental adaptability of the disease vectors. Specifically, there is a lack of studies that combine animal outbreak data with ecological and climatic variables to predict where disease-carrying vectors are likely to live and spread the disease.

Ecological niche modeling (ENM) offers a robust framework for incorporating environmental and ecological factors into predictive spatial analyses [21]. Among ENM techniques, the maximum-entropy model (MaxEnt) is particularly well suited for this purpose, as it allows for accurate predictions using presence-only data and a suite of environmental variables [22,23]. By analyzing the environmental conditions of previously affected locations, the model highlights other regions with similar profiles that could potentially experience the disease [24]. Previous studies have applied models to predict the potential distribution of vector-borne animal diseases such as lumpy skin disease and foot-and-mouth disease and their associated vectors in various regions [24,25,26,27]. Previous studies in Iran, Russia, and China have successfully applied MaxEnt to map the ecological suitability of LSD vectors under diverse environmental conditions. These applications demonstrate the flexibility and consistency of the method in predicting spatial risk patterns, making it a relevant and appropriate choice for assessing LSD risk across the varied landscapes of Thailand [24,28,29]. These investigations have highlighted the utility of the model in identifying high-risk areas, elucidating the ecological requirements of the vectors, and supporting preparedness efforts.

Despite the advancements in predictive ecological modeling, the application of maximum-entropy modeling to LSD in Thailand remains underexplored. Consequently, there is a critical need to investigate its utility in this context to enhance understanding of vector ecology and refine disease control planning. Therefore, this study aimed to model the environmental suitability and spatial risk distribution of LSD in Thailand using maximum-entropy modeling. The resulting risk maps provide a foundation to inform and prioritize targeted surveillance and control measures.

## 2. Materials and Methods

### 2.1. Study Area

Thailand covers approximately 513,115 km^2^, extending between latitudes 5°37′ N to 20°27′ N and longitudes 97°22′ E to 105°37′ E. The country is characterized by a variety of landscapes, including the mountainous regions in the north, the fertile central plains predominantly shaped by the Chao Phraya River, the semi-arid Khorat Plateau in the northeast, and the coastal lowlands in the south. These diverse geographic features contribute to distinct ecological zones and vector habitats. The climate of Thailand is significantly affected by the southwest and northeast monsoons, resulting in three primary seasons: hot (March–May), rainy (mid-May to October), and cool (November–February) [30]. Insect vectors of LSD, such as stable flies and mosquitoes, are typically present on cattle farms throughout the year [31,32,33,34].

For all mapping, this study used the Thai administrative boundary shapefile, obtained from the public website https://data.humdata.org/dataset/cod-ab-tha (accessed on 18 March 2025).

### 2.2. Lumpy Skin Disease Data

This study utilized LSD outbreak data collected across Thailand, provided by the Department of Livestock Development (DLD), Thailand. These outbreak records, which were officially submitted to the World Organisation for Animal Health (WOAH), are accessible via the World Animal Health Information System (WAHIS) at https://wahis.woah.org (accessed on 18 March 2024). A total of 669 reported outbreaks of LSD were included in this analysis. The data covering the period from 2021 to 2023 included outbreak details such as onset date, geographic coordinates (latitude and longitude), and the number of infected animals.

Because outbreak reports in 2022–2023 were sparse compared with the widespread epidemic in 2021, the analysis examined whether including these lower-incidence years would meaningfully alter model outcomes. The rationale was that, 2021, as the initial epidemic year, likely captured the widest geographic spread and the broadest range of environmental conditions associated with LSD occurrence, whereas the later years recorded a very small number of outbreaks. To assess this, the data were divided into two subsets: LSD outbreaks from 2021 to 2023 (LSD 2021–2023 dataset) and outbreaks from 2021 only (LSD 2021 dataset). The primary analysis was conducted utilizing the combined 2021–2023 dataset, and the results were subsequently compared with those from the 2021-only dataset to further investigate potential differences.

### 2.3. Predictor Variable Data Collection and Processing

In developing the model, a total of 22 variables were selected based on their potential biological relevance to the distribution of LSD vectors across Thailand (Table 1). The spatial distributions of the predictor variables are illustrated in Figure 1. In each map, the color gradient represents the range of values for each specific variable across the country, where darker or more-intense shades generally indicate higher values, and lighter shades indicate lower values (Figure 1).

The natural variables included 19 bioclimatic factors (Bio1–Bio19), which reflect global climate patterns and were obtained from the WorldClim database (https://worldclim.org/data; accessed on 14 April 2025). These variables are commonly utilized in spatial modeling studies related to animal diseases [27,35,36].

Additionally, this study incorporated environmental variables, including the normalized difference vegetation index (NDVI) and land cover type 1. NDVI data were retrieved from MODIS MOD13A1 (https://ladsweb.modaps.eosdis.nasa.gov/missions-and-measurements/products/MOD13A1; accessed on 14 April 2025), while land cover information was obtained from MODIS MCD12Q1 (https://ladsweb.modaps.eosdis.nasa.gov/missions-and-measurements/products/MCD12Q1/; accessed on 14 April 2025).

Furthermore, cattle density data were sourced from the Food and Agriculture Organization (FAO) global livestock distribution database (https://www.fao.org; accessed on 14 April 2025). The Google Earth Engine (GEE) was employed to define the study area and to apply an algorithm for computing spatial values at the pixel level. All variables were subsequently exported in comma-separated values (CSVs) and tagged image file format (TIFF) formats.

### 2.4. Variable Selection

Prior to model development, a multicollinearity test was performed to prevent overfitting, utilizing Pearson correlation analysis in R software (version 2023.12.0+369). Variables exhibiting a correlation coefficient greater than 0.8 were considered highly correlated. In such cases, only the variable with greater biological plausibility was retained for further modeling, while the correlated counterpart was excluded from subsequent analysis [37,38]. Following the multicollinearity assessment, seven variables were selected for inclusion in the final model. The selected variables were subsequently converted to American Standard Code for Information Interchange (ASCII) format using QGIS version 3.34.2-Prizren. These predictors were used as input in the model algorithm, which generated spatial predictions based on outbreak data.

### 2.5. Modeling Using MaxEnt

The model estimates the most uniform probability distribution constrained by the empirical averages of environmental variables at known presence locations. It aims to characterize the spatial distribution of occurrences based on these constraints. The mathematical formulation is as follows [39,40]:$$P_{\omega }(y|x)=\frac{1}{Z_{\omega }(x)}\exp\left(\sum_{i=1}^{n}\omega_{i}f_{i}(x,y)\right)$$$$Z_{w}\left(x\right)=\sum_{y}\exp\left(\sum_{i=1}^{n}\omega_{i}f_{i}\left(x,y\right)\right)$$
where x denotes the environment variable input into the model; y represents the predicted area; $f_{i}(x,y)$ denotes the feature functions corresponding to environmental variables. The term $\omega_{i}$ refers to the weight associated with $f_{i}(x,y)$. The constant $Z_{\omega }\left(x\right)$ is a normalization factor.

The maximum-entropy modeling was performed using MaxEnt software version 3.4.4. To ensure robust validation, the occurrence data were partitioned into two subsets, with 80% allocated for model training and the remaining 20% reserved for testing [26]. Furthermore, the jackknife test was utilized to evaluate the contribution of each environmental variable by measuring the training gain when the model is run using that variable alone, providing insight into its individual predictive power [41].

### 2.6. Parameter Setting

The following options were selected in the settings panel: give visual warnings, show tooltips, ask before writing, remove duplicate presence records, write clamp grid when projecting, perform MESS analysis when projecting, add samples to background, add all samples to background, write plot data, extrapolate, perform clamping, write output grids, write plots, cache ASCII files, log-scale raw/cumulative pictures, and 500 maximum iterations. In addition, the model utilized 10,000 randomly selected background points as pseudo-absence data. To reduce the risk of overfitting, a regularization multiplier of 1 was applied, and the model was replicated 10 times [26,42].

### 2.7. Model Performance Evaluation

The model’s performance was assessed through jackknife tests, response curves, and cross-validation, with area under the curve (AUC) used as a key evaluation metric. The AUC values range from 0 to 1, where scores > 0.5 indicate predictive ability greater than random chance [43], and higher values reflect stronger model accuracy. In addition, the resulting risk map with values scaled between 0 and 1 illustrates the spatial gradient of disease occurrence probability, from low- to high-risk areas.

In addition, for the model with a higher AUC, model performance was assessed using the True Skill Statistic (TSS) [44] and omission rate [45]. The TSS was calculated for each replicate using the maximum training sensitivity plus specificity threshold reported by MaxEnt, with sensitivity and specificity derived from the predicted cloglog values for presence points and background points. The omission rate was extracted directly from the MaxEnt output for the same threshold. Mean values across all replicates were used to summarize the model performance.

The summary of the workflow for variable selection, modeling, model evaluation, and risk map generation is shown in Figure 2.

## 3. Results

### 3.1. Model Performance and Variables Used in the Final Model

The performance of the model developed to predict the potential distribution of LSD in Thailand from 2021 to 2023 was evaluated using the receiver operating characteristic (ROC) curve (Figure 3). The model achieved a mean AUC value of 0.699 (~0.70), indicating a moderate predictive capability. This AUC value demonstrates that the model was able to distinguish between suitable and unsuitable habitats for LSD occurrence with reasonable accuracy. The variable with the highest contribution was land cover, and the lowest contribution was from annual precipitation (Table 2). In addition, the analysis using the LSD 2021 dataset yielded a slightly higher AUC value of 0.708, indicating moderate model performance (Appendix A). Based on the maximum training sensitivity plus specificity threshold, the model achieved a sensitivity of 0.778 ± 0.042 and a specificity of 0.557 ± 0.041, yielding a mean TSS of 0.335 ± 0.004 across ten replicates with minimal variation (0.328–0.342). Omission rates averaged 0.215 for the training data and 0.274 for the test data, indicating a balanced trade-off between detecting presence points and limiting overprediction and confirming the reliable identification of environmentally suitable areas for LSD occurrence.

### 3.2. Response Curves for Important Variables in the LSD Model

The response curves of the model highlight the contribution of individual environmental variables to the predicted probability of LSD presence in Thailand, as illustrated in Figure 4. These curves reflect how changes in each parameter influence the model prediction. The NDVI demonstrated a non-linear positive association with LSD risk, with the highest predicted probability occurring in areas with moderate levels of vegetation. The highest predicted suitability based on land cover was found in Class 13 (urban and built-up lands).

Cattle density showed a clear positive relationship with LSD suitability, where areas with a higher livestock density corresponded to a greater probability of disease presence. Among the bioclimatic variables, precipitation of the wettest quarter showed a positive association with LSD presence, where increasing precipitation enhanced the suitability of the disease. Annual precipitation was linked to a gradual decline in predicted suitability as rainfall increased, suggesting that overly wet environments may be less conducive to disease occurrence. Precipitation of the driest month exhibited a sharp reduction in suitability under wetter dry-season conditions. Meanwhile, the mean diurnal temperature range showed higher suitability in areas with a moderate daily temperature variation. In addition, the results obtained from the 2021 dataset showed a consistent outcome when compared with the full dataset (Appendix A).

### 3.3. The Contribution of Each Environmental Variable to the Model

The jackknife test was conducted to assess the contribution of each environmental variable to the LSD model (Figure 5). Land cover demonstrated the highest predictive contribution when analyzed independently, indicating its status as the most informative variable in the dataset. NDVI and cattle density also exhibited significant individual effects, supporting their importance in the model. Furthermore, the bioclimatic variables including mean diurnal range, annual precipitation, precipitation of the driest month, and precipitation of the wettest quarter also influenced the model. Notably, the analysis based solely on the LSD 2021 dataset yielded results consistent with those derived from the comprehensive LSD 2021–2023 dataset (Appendix A).

### 3.4. Potential Risk Map of Lumpy Skin Disease

The map illustrating the predicted suitability for LSD distribution across Thailand is presented in Figure 6. This map displays a continuous probability scale, ranging from low suitability (represented in green) to high suitability (indicated in red), which reflects the likelihood of LSD presence based on current environmental conditions. The model identifies extensive regions of high suitability for LSD occurrence, particularly in the central and northeastern areas, which are depicted in orange-to-red shades. Moderate suitability is noted in certain regions of the southern provinces, while areas with low suitability, represented in green, are located in the northern, western, and some parts of the southern regions.

## 4. Discussion

This study utilized ecological niche modeling to assess the potential distribution of LSD occurrences in Thailand, based on environmental and climatic variables related to spatial suitability. The analysis identified land cover type 1, cattle density, and NDVI as the most significant factors influencing LSD distribution. Furthermore, precipitation metrics, specifically annual precipitation, precipitation of the driest month, and precipitation of the wettest quarter, along with the mean diurnal range, also exerted an impact on the model.

LSDV is primarily transmitted among susceptible hosts by blood-feeding arthropod vectors, including various species of mosquitoes, biting flies, and ticks [46,47]. The co-occurrence of competent vectors and available hosts can facilitate conditions that support virus transmission and increase the likelihood of disease spread. In the present study, land cover was considered as the most influential predictor of LSD occurrence. This finding is consistent with previous studies that have highlighted the role of land cover in determining areas at risk [26,48]. In this study, land classified as urban and built-up areas (MODIS Class 13) contributed most to the model. This result likely reflects the fact that, in Thailand, a substantial number of cattle farms, particularly those operated by smallholders and backyard producers, are located within or adjacent to urban or built-up areas [49]. The presence of more cattle in these settings provides a greater number of susceptible hosts for the LSDV, increasing the risk of disease emergence and transmission in these areas. In addition to host presence, urban and peri-urban environments often exhibit warmer microclimates that may enhance the survival, reproduction, and activity of vector species, thereby increasing the risk of vector-borne disease transmission [48]. This interpretation is also consistent with the finding that areas with sparse or absent cattle populations were not identified as suitable for LSD occurrence, likely due to the limited availability of hosts required to sustain transmission cycles.

In the present study, cattle density is determined as an important factor contributing to the risk of LSD, indicating that regions with higher livestock concentrations are more vulnerable to outbreaks. This observation is consistent with previous findings, such as a study conducted in West Kazakhstan, which identified cattle density as a significant risk factor for LSD occurrence [50]. Areas with high cattle density likely provide stable flies with greater feeding opportunities and consistent access to hosts, supporting the development and maintenance of larger vector populations. This ecological association may enhance vector persistence and increase the risk of mechanical transmission of LSDV, particularly in settings in which vector control measures are limited or inconsistently applied [51].

The NDVI serves as a key indicator of vegetation health and density, which can indirectly reflect habitat suitability for vectors. Higher NDVI values typically correspond to areas with dense vegetation that provide breeding sites, resting areas, and favorable microclimatic conditions for vector persistence [52,53]. In the context of the LSDV, such environments may support increased vector abundance and activity, thereby elevating the likelihood of virus transmission among susceptible cattle populations. This interpretation is supported by previous studies that have shown that landscape features, including vegetation patterns measured by NDVI, can influence vector abundance. Areas with a moderate vegetation density may offer favorable conditions for vector survival and reproduction, thereby contributing to the spatial distribution of LSD risk [51,54].

Furthermore, studies reveal that temperatures between 25 °C and 32 °C and relative humidity levels above 60% can enhance vector survival and reproduction [55,56], thereby increasing the potential for disease transmission. Previous studies have also shown that LSD outbreaks often coincide with seasons or regions in which vector populations are thriving [57,58], suggesting a strong ecological association between environmental suitability and disease risk. Favorable habitats can sustain larger vector populations, which not only increases the likelihood of within-herd transmission but also facilitates the spread of LSDV between herds across broader geographic areas. In this study the model also highlighted the importance of one temperature variable, namely, the mean diurnal range and three precipitation variables including the annual precipitation, precipitation of the driest month, and precipitation during the wettest quarter. These results complement broader patterns observed in LSDV epidemiology and highlight the multifactorial nature of disease transmission [59,60].

Collectively, key environmental and climatic predictors, such as isothermality, mean temperature of the wettest quarter, annual and seasonal precipitation, NDVI, and land cover, emphasize the importance of temperature stability, vegetation density, and habitat structure in supporting vector populations. Moreover, high cattle density and the dominance of smallholder and backyard farming systems near urban and peri-urban areas may further intensify exposure to vectors, particularly in settings in which biosecurity is limited, and animal movement is poorly regulated. Additionally, increased insect activity has been associated with warm temperatures, high moisture levels, and land cover types that provide ideal breeding and resting environments [61]. Taken together, these interacting factors help explain the spatial patterns of disease transmission, with such risk-favorable conditions present in several regions of Thailand [62].

The minimal difference in AUC between the 2021–2023 and 2021 datasets indicates that data from the initial outbreak year alone were sufficient to capture the core spatial patterns of LSD distribution. This is likely because 2021 represented both the widest geographic spread and the highest transmission intensity, encompassing the environmental and management-related conditions most relevant to disease occurrence [15]. Subsequent years, which recorded fewer and more spatially clustered cases, contributed less new spatial information, explaining the comparable performance across scenarios. The mean AUC values (~0.70) fall within the range generally interpreted as moderate discrimination ability in ecological niche modeling. Additionally, the mean TSS, in combination with strong sensitivity and reasonable specificity, further reflects a moderate ability to discriminate between suitable and unsuitable areas for LSD. The low variation in TSS across replicates demonstrates consistent performance, while omission rates show that most presence points were correctly identified, with some extension of predicted suitability beyond observed occurrences.

The risk map indicates high suitability for LSD occurrences in central and northeastern Thailand, which closely reflects previously observed outbreak patterns. Earlier studies have shown that an LSD outbreak was first reported in the northeastern region and gradually expanded into other parts of the country, with a clear concentration of outbreaks in the central and northeastern provinces [14,15]. The close alignment between model predictions and actual outbreak locations supports the validity of the model and emphasizes its potential as a tool for guiding disease monitoring and early intervention strategies. In addition, the high-risk classification of central and northeastern Thailand may be attributed to a convergence of environmental and livestock management factors. Environmentally, these regions experience moderate-to-high temperatures and relatively stable humidity, conditions that are highly conducive to the survival, reproduction and activity of mechanical vectors that are suspected to play a role in the transmission of LSD. From a livestock management perspective, these regions are characterized by high cattle density and widespread traditional or semi-intensive farming systems [63]. The combination of conducive environmental conditions and livestock management practices creates a favorable epidemiological setting for the persistence and spread of LSD, thereby justifying the elevated risk classification observed in the model outputs.

The results of the model demonstrate a strong alignment with the ecological characteristics of LSD and its known vectors, which reinforces the credibility of the spatial predictions produced. These insights are particularly useful for decision-makers and professionals in veterinary and public health sectors. The risk maps and variable analyses can inform more-strategic surveillance, focused vector control efforts, and more-efficient allocation of resources in areas identified as high risk. Furthermore, integrating the model outputs into operational planning and early warning systems can strengthen preparedness and support timely and coordinated responses to future outbreaks.

The findings from this study have important implications for guiding targeted control strategies. Interrupting the transmission cycle remains one of the most effective strategies for the prevention and control of infectious diseases. In the case of LSD in Thailand, vector control has been identified as a critical factor influencing disease transmission [64]. Both the environmental conditions and vector habitat suitability in Thailand are highly conducive to the establishment and spread of LSD, highlighting the need for proactive intervention. To mitigate the risk of future outbreaks, a comprehensive approach should be adopted that includes regular cattle vaccination, strict control of animal movement, enhanced vector monitoring, and continuous disease surveillance. Sustaining these measures is essential to safeguard national cattle health and prevent re-introduction of the disease. Although no outbreaks have been reported in Thailand in 2025 (as of 15 August 2025), maintaining these control measures remains critical, as the country has not yet been officially recognized as free from the disease.

To the best of our knowledge, this is the first study to apply ecological niche modeling to predict the potential distribution of LSD occurrences in Thailand. While the study provides valuable insights into spatial risk patterns, several limitations must be acknowledged. The model was constructed using presence-only outbreak data due to the lack of detailed entomological information, thereby requiring the assumption of a uniform distribution of vectors across the country. Nevertheless, MaxEnt remains a well-suited tool for presence-only modeling and has shown consistent reliability under such constraints [65]. Additionally, the absence of data has necessitated the exclusion of potentially significant variables, such as animal movement patterns and insect abundance [66]. Notably, these variables have also been unavailable in several previous studies on the epidemiology of LSD, reflecting a common limitation within the existing body of research. Future research should incorporate these and other relevant variables to strengthen the robustness of the findings.

## 5. Conclusions

This study successfully applied ecological niche modeling to predict high-risk areas for LSD in Thailand and identified key variables influencing its spatial suitability. High-risk areas in central and northeastern Thailand closely matched historical outbreak data, supporting the model’s reliability. These findings provide a valuable basis for targeted surveillance, resource allocation, and early warning efforts. Integrating such spatial predictions into disease control planning can enhance preparedness and improve response strategies against future LSD outbreaks.

## Figures and Tables

**Figure 1 animals-15-02456-f001:**
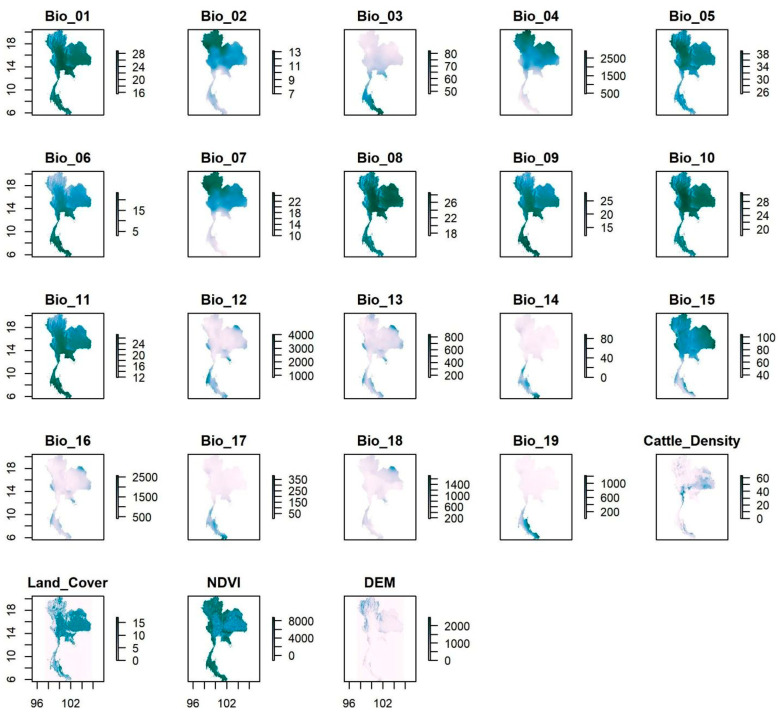
Spatial distribution of predictor variables for the lumpy skin disease model in Thailand. Each panel shows the spatial distribution of a variable across Thailand. The x- and y-axes represent longitude and latitude, respectively. The scale bars on the right indicate the variable-specific values, which correspond to the ranges summarized in Table 1. Darker shades indicate higher values, while lighter shades indicate lower values.

**Figure 2 animals-15-02456-f002:**
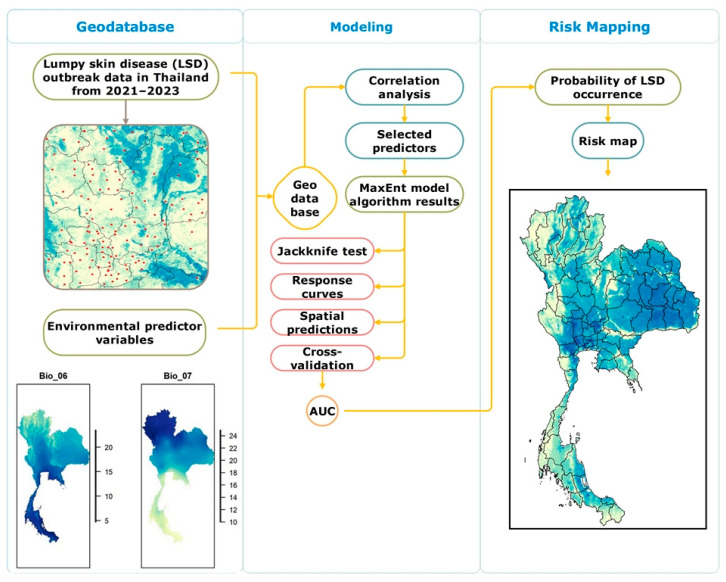
Modeling workflow for lumpy skin disease (LSD) model. The process includes data compilation, predictor selection, model evaluation including jackknife test, response curves and area under the curve (AUC), and generation of a risk map showing the probability of LSD occurrence.

**Figure 3 animals-15-02456-f003:**
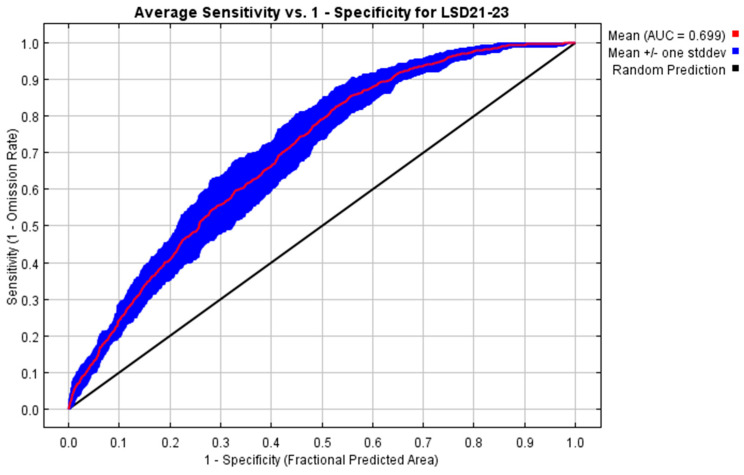
Model performance for the lumpy skin disease model using the LSD 2021–2023 dataset. The area under the curve (AUC = 0.699) reflects moderate model performance.

**Figure 4 animals-15-02456-f004:**
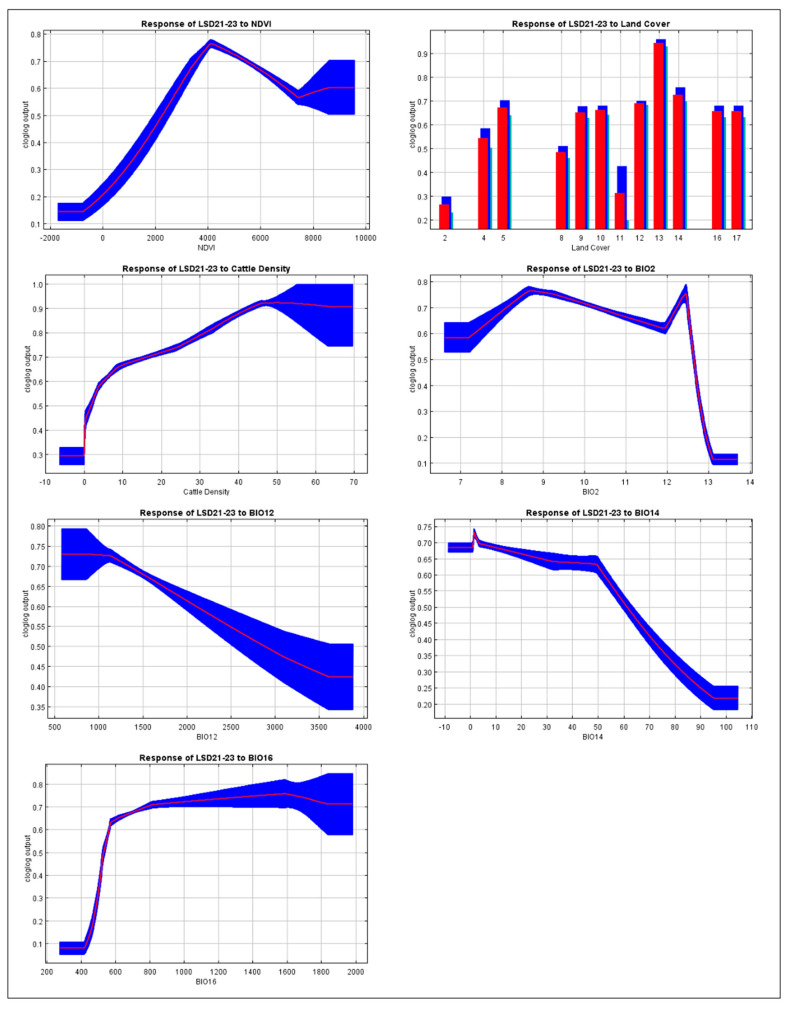
Response curves of environmental variables influencing the maximum-entropy model using the LSD 2021–2023 dataset. The model shows the effect of each variable on disease suitability. Red lines represent mean predicted probability and blue areas indicate variation among replicates. In the bar graph, red bars denote mean probability and blue bars represent variation across replicates.

**Figure 5 animals-15-02456-f005:**
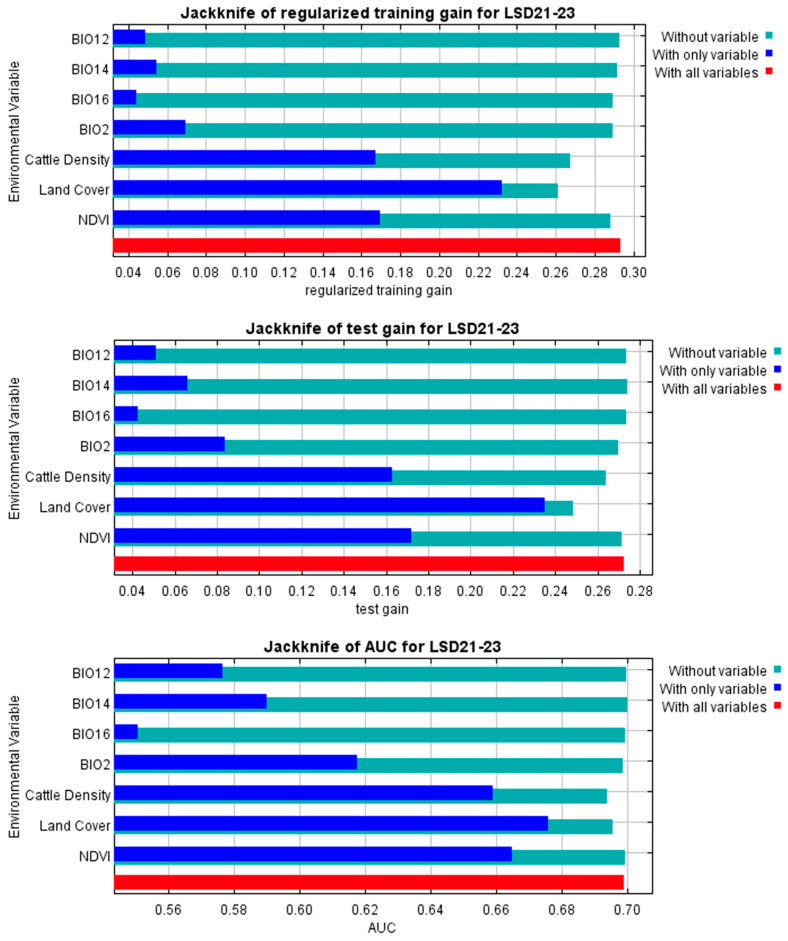
The jackknife test for evaluating the relative importance of environmental variables using the LSD 2021–2023 dataset: Land cover contributed the most to the model, followed by normalized difference vegetation index and cattle density with bioclimatic variables also contributing to the model.

**Figure 6 animals-15-02456-f006:**
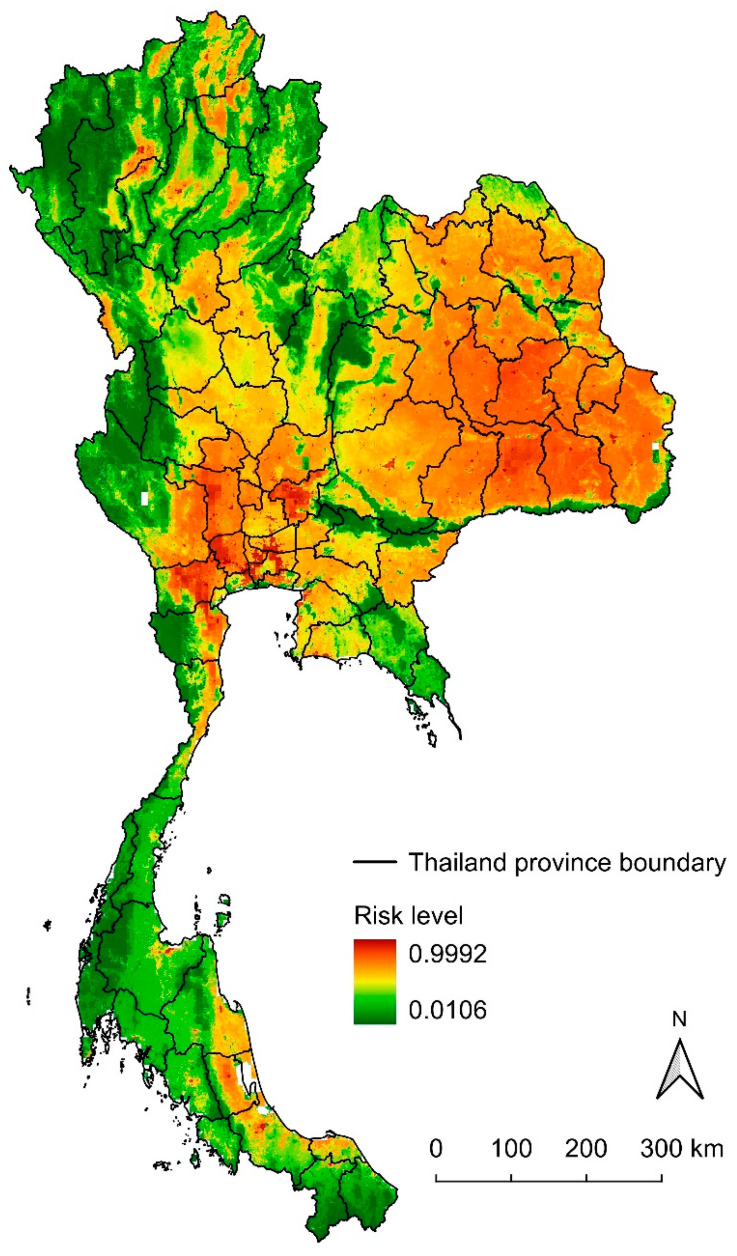
The geographic distribution of the potential risk of LSD in Thailand. Warmer colors indicate higher suitability while cooler colors represent lower suitability. The Thailand administrative boundary shapefile (map) was downloaded from a public website (https://data.humdata.org/dataset/cod-ab-tha; accessed on 18 March 2025).

**Table 1 animals-15-02456-t001:** Environmental and climatic variables considered for lumpy skin disease model.

Variable	Description	Source	Included
Bio 1	Annual mean temperature	WorldClim	No
Bio 2	Mean diurnal range	WorldClim	Yes
Bio 3	Isothermality	WorldClim	No
Bio 4	Temperature seasonality	WorldClim	No
Bio 5	Max temperature of the warmest month	WorldClim	No
Bio 6	Min temperature of the coldest month	WorldClim	No
Bio 7	Temperature annual range	WorldClim	No
Bio 8	Mean temperature of the wettest quarter	WorldClim	No
Bio 9	Mean temperature of the driest quarter	WorldClim	No
Bio 10	Mean temperature of the warmest quarter	WorldClim	No
Bio 11	Mean temperature of the coldest quarter	WorldClim	No
Bio 12	Annual precipitation	WorldClim	Yes
Bio 13	Precipitation of the wettest month	WorldClim	No
Bio 14	Precipitation of the driest month	WorldClim	Yes
Bio 15	Precipitation seasonality	WorldClim	No
Bio 16	Precipitation of the wettest quarter	WorldClim	Yes
Bio 17	Precipitation of the driest quarter	WorldClim	No
Bio 18	Precipitation of the warmest quarter	WorldClim	No
Bio 19	Precipitation of the coldest quarter	WorldClim	No
Cattle density	The density of cattle	FAO	Yes
NDVI	Vegetation density and greenness	MODIS MOD13A1	Yes
Land cover type 1 ^1^	IGBP global vegetation classification scheme	MODIS MCD12Q1	Yes

^1^ Land cover type 1 data include 17 land cover classes.

**Table 2 animals-15-02456-t002:** Variables used in final models and variable contribution.

Variable	Description	Variable Contribution (%)
Bio 2	Mean diurnal range	1.6
Bio 12	Annual precipitation	0.6
Bio 14	Precipitation of the driest month	1
Bio 16	Precipitation of the wettest quarter	4
Cattle density	The density of cattle	25
NDVI	Vegetation density and greenness	2.8
Land cover type 1	IGBP global vegetation classification scheme	65

## Data Availability

The data used in this study are available from the World Animal Health Information System (WAHIS) at https://wahis.woah.org (accessed on 18 March 2024).

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
