# Peer review of "Spatial Risk Distribution of Lumpy Skin Disease in Thailand Based on Maximum-Entropy Modeling"

_animals, 2025, doi:10.3390/ani15162456_

Round 1

Reviewer 1 Report

Comments and Suggestions for Authors

In my perspective from the point of view of methodology of modelling the manuscript looks very good and have no issue with it, or, nay be my knowledge in this technical area is not enough to judge it.

From the epidemiological point of view this model looks like natural scenario: it takes into account the same point of entry of infection as it was during last outbreak, the virus will spread over naive population of cattle not taking into account the herd immunity, it will be no long distance jump of the disease for the reason of human activity and animal movement.

Also from the manuscript it is not clear the status of the Thailand in 2025: free from LSD with vaccination?

What are the climateу conditions in Thailand and how is the seasonality present in the country? Is there any vector free period and the term "overwintering" applicable? I do not think that most of the potential readers well-known with this information.

Also using words "warmer", "higher" do not make the understanding easier.

Furthermore, studies reveal that warmer temperatures and higher humidity levels 319
can enhance vector survival and reproduction 

From the paper we can find that previous outbreak lasted from 2021 to 2023. Probably more information about this event will help better understand the specific risk factors for Thailand.

What is the situation with LSD in neighboring countries and what is potential source of virus introduction to the country? Do authors taking into account this information?

Again, the modelling looks nice but the scenario looks ideal and for this reason not feasible. 

Author Response

Reviewer 1

Response: Overall, we sincerely thank the reviewer for painstakingly evaluated of our manuscript and have provided constructive comments. We are truly grateful for the time and effort dedicated to reviewing our work and for the valuable feedback and suggestions rendered to us.

In my perspective from the point of view of methodology of modelling the manuscript looks very good and have no issue with it, or, nay be my knowledge in this technical area is not enough to judge it.

From the epidemiological point of view this model looks like natural scenario: it takes into account the same point of entry of infection as it was during last outbreak, the virus will spread over naive population of cattle not taking into account the herd immunity, it will be no long-distance jump of the disease for the reason of human activity and animal movement.

Response: We appreciate the reviewer’s interest in potential sources of LSD introduction and agree that transboundary animal movement is an important pathway for disease spread. However, the present study applies the MaxEnt framework as an ecological niche model to evaluate the relationship between LSD occurrence and environmental variables, with the aim of producing a spatial risk surface indicating areas with higher environmental suitability for LSD occurrence within Thailand.

MaxEnt is specifically designed to model presence–background data against environmental predictors, and does not directly incorporate dynamic anthropogenic factors such as animal movement or trade flows. While these are recognized as critical drivers of transboundary spread, incorporating them into the present modelling framework is not feasible because animal movement data at the national or cross-border scale are not publicly available at the required spatial and temporal resolution, and movement volumes change daily.

Even in previous LSD risk mapping studies in Asia (Li et al., 2023) and other transboundary disease modelling using MaxEnt such as Foot and Mouth disease (An et al., 2025; Gunasekera et al., 2025), animal movement variables are absent due to data limitations, with the modelling focusing instead on environmental suitability (e.g., climate, host density, land cover).

Therefore, the “scenario” presented by the MaxEnt model should not be interpreted as a complete transmission model, but rather as a prediction of environmental suitability for LSD occurrence under the available environmental conditions. This information complements, rather than replaces, epidemiological analyses of transmission routes and can help guide targeted surveillance and prevention efforts, including in areas near borders where environmental conditions are suitable for LSD but where movement data are lacking.

This study is also a follow up study as we have addressed risk factors for LSD introduction to Thailand (Arjkumpa et al., 2021) and risk factors (Arjkumpa et al., 2024) for the disease spread across the country.

Also, from the manuscript it is not clear the status of the Thailand in 2025: free from LSD with vaccination?

Response: Based on WOAH–WAHIS data, no LSD outbreaks have been reported in Thailand in 2025 (accessed 15 August 2025). This information has been added to the revised manuscript (lines 412–414). While vaccination has been implemented in Thailand, the absence of reported outbreaks alone does not confirm LSD-free status under vaccination at this current stage.

What are the climateу conditions in Thailand and how is the seasonality present in the country? Is there any vector free period and the term "overwintering" applicable? I do not think that most of the potential readers well-known with this information.

Response: We appreciate this insightful comment and agree that providing more detailed climatic context is important for readers unfamiliar with the region. In the revised manuscript, the “Study Area” section has been expanded to include a clearer description of Thailand’s climate (see line 107-109). In addition, we have noted that insect vectors such as stable flies and mosquitoes are typically abundant on cattle farms year-round in Thailand, supported by relevant research references (line 109-111). Consequently, there is no distinct vector-free period, and the concept of “overwintering” is not applicable in the tropical climate of Thailand.

Also using words "warmer", "higher" do not make the understanding easier.

Furthermore, studies reveal that warmer temperatures and higher humidity levels 319
can enhance vector survival and reproduction 

Response: Thank you for your insight. We added specific temperature and humidity thresholds from referenced studies to improve clarity and support better understanding (line 340-341)

From the paper we can find that previous outbreak lasted from 2021 to 2023. Probably more information about this event will help better understand the specific risk factors for Thailand.

Response: Thank you for this helpful comment. We agree that providing more contextual detail about the 2021–2023 outbreak period strengthens the epidemiological interpretation of our findings. In response, we have added a brief summary of key features to the introduction. The following sentence has been added to the manuscript: “LSD has emerged as a significant transboundary disease affecting cattle populations in Thailand since its initial detection in 2021, with animal movement from previously affected countries hypothesized as the primary source of introduction. The disease rapidly expanded into central regions and, within a few months, spread across most parts of the country. This pattern of spread was consistent with known risk factors, including smallholder farming systems and limited vector control measures” (line 59-64).

What is the situation with LSD in neighboring countries and what is potential source of virus introduction to the country? Do authors taking into account this information?

Response: The potential risk factor for LSD virus introduction into Thailand and the situation in neighboring countries have been addressed in our previous publication (Arjkumpa et al., 2021; Arjkumpa et al., 2024). This information has now been incorporated into the revised manuscript at line 59-61 to provide additional context for the present study. Furthermore, given that the model requires environmental surface data across Thailand such as landcover, NDVI and temperature, the data of source of virus is not fit for the analysis. Please see the following response for detailed explanation.

Again, the modelling looks nice but the scenario looks ideal and for this reason not feasible. 

Response: We thank the reviewer for the comment. In this study, MaxEnt was applied as an ecological niche model to estimate the environmental suitability for LSD occurrence within Thailand. The objective was to identify areas with environmental conditions favorable for LSD, not to reconstruct all potential transmission pathways. MaxEnt relies on environmental predictors and presence–background data, and does not directly incorporate dynamic anthropogenic factors such as animal movement. These movements vary daily, are influenced by informal trade, and lack publicly available national or cross-border datasets at the spatial and temporal resolution required. This limitation is consistent with previous LSD MaxEnt studies, which also omit such variables for the same reason.

Numerous publications have successfully applied ecological niche modeling to a wide range of animal and human diseases, establishing it as a robust and well-recognized approach in epidemiological research. This method has proven particularly valuable for generating suitability risk maps, which help identify areas at higher risk based on environmental and ecological factors. We believe that incorporating this approach is highly relevant to advancing the field, as it provides critical insights that are difficult to obtain through traditional methods alone. Furthermore, the extensive body of literature demonstrates not only the feasibility of conducting ecological niche modeling but also its practical utility in guiding surveillance, prevention, and control strategies at both national and regional levels. For example, ecological niche modeling has been applied to map high-risk areas for vector-borne and transboundary diseases such as lumpy skin disease in China and Russia (Li et al., 2023; Sprygin et al., 2020), foot-and-mouth disease in Turkey (Samy & Peterson, 2016), dengue in Southeast Asia (Souto et al., 2022), and malaria in Africa (Moffett et al., 2007). These examples demonstrate the method’s versatility and its increasing acceptance in both veterinary and public health disciplines.

References:

An Q, Lv Y, Li Y, Sun Z, Gao X, Wang H. Global foot-and-mouth disease risk assessment based on multiple spatial analysis and ecological niche model. Vet Q. 2025 Dec;45(1):1-11. doi: 10.1080/01652176.2025.2454482. Epub 2025 Jan 21. PMID: 39838825; PMCID: PMC11755741.

Arjkumpa O, Suwannaboon M, Boonrod M, Punyawan I, Liangchaisiri S, Laobannue P, Lapchareonwong C, Sansri C, Kuatako N, Panyasomboonying P, Uttarak P, Buamithup N, Sansamur C, Punyapornwithaya V. The First Lumpy Skin Disease Outbreak in Thailand (2021): Epidemiological Features and Spatio-Temporal Analysis. Front Vet Sci. 2022 Jan 7;8:799065. doi: 10.3389/fvets.2021.799065. PMID: 35071388; PMCID: PMC8782428.

Arjkumpa O, Wachoom W, Puyati B, Jindajang S, Suwannaboon M, Premashthira S, Prarakamawongsa T, Dejyong T, Sansamur C, Salvador R, Jainonthee C, Punyapornwithaya V. Analysis of factors associated with the first lumpy skin disease outbreaks in naïve cattle herds in different regions of Thailand. Front Vet Sci. 2024 Feb 22;11:1338713. doi: 10.3389/fvets.2024.1338713. PMID: 38464702; PMCID: PMC10921558.

Gunasekera U, Alkhamis MA, Puvanendiran S, Das M, Kumarawadu PL, Sultana M, Hossain MA, Arzt J, Perez A. Ecological niche modeling for surveillance of foot-and-mouth disease in South Asia. PLoS One. 2025 Apr 22;20(4):e0320921. doi: 10.1371/journal.pone.0320921. PMID: 40261938; PMCID: PMC12013921.

Moffett A, Shackelford N, Sarkar S. Malaria in Africa: vector species’ niche models and relative risk maps. PLoS One. 2007;2(9):e824.

Samy AM, Peterson AT. Climate change influences on the global potential distribution of foot-and-mouth disease virus. PLoS One. 2016;11(11):e0164681.

Souto RN, Ferreira AF, de Sá ILR, Rocha DD, de Almeida SS, Rivas GGB, et al. Ecological niche models for dengue in Southeast Asia: implications for surveillance and control. Acta Trop. 2022;232:106478.

Sprygin A, Babin Y, Pestova Y, Kononov A, Wallace DB. Ecological niche modeling of lumpy skin disease virus in Russia. Front Vet Sci. 2020;7:583706.

  1. Li, Q. An, Z. Sun, X. Gao, and H. Wang, “Risk Factors and Spatiotemporal Distribution of Lumpy Skin Disease Occurrence in the Asian Continent During 2012-2022: An Ecological Niche Model,” Transboundary and Emerging Diseases. (2023): 6207149, 10

Reviewer 2 Report

Comments and Suggestions for Authors

In the article, “Spatial Risk Distribution of Lumpy Skin Disease in Thailand Based on the Maximum Entropy Modeling,” the authors present a study aimed to evaluate the spatial distribution of LSD risk in Thailand using a maximum entropy modeling in contests of supporting surveillance and control efforts for LSD in the country

Here are my comments:

Martials and Methods

2.1. Study area

This information is not proper for the section, may be will be better to be in Discussion

Lines 260-270 - There is mistake in description of the colours.

There is a discrepancy in the statements:

Lines 224 – 226 – “The NDVI demonstrated a non-linear positive association with LSD risk, with the highest predicted probability occurring in areas with moderate levels of vegetation.”

Line 306-308 – “Areas with greater vegetation density may offer favorable conditions for vector survival and reproduction, thereby contributing to the spatial distribution of LSD risk [25, 44-45].”

What is your opinion?

Author Response

Reviewer 2

In the article, “Spatial Risk Distribution of Lumpy Skin Disease in Thailand Based on the Maximum Entropy Modeling,” the authors present a study aimed to evaluate the spatial distribution of LSD risk in Thailand using a maximum entropy modeling in contests of supporting surveillance and control efforts for LSD in the country.

 Response: Overall, we sincerely thank the reviewer for painstakingly evaluated of our manuscript and have provided constructive comments. We are truly grateful for the time and effort dedicated to reviewing our work and for the valuable feedback and suggestions rendered to us.

Here are my comments:

Martials and Methods

2.1. Study area

This information is not proper for the section, may be will be better to be in Discussion

Response: Thank you for this comment. While we understand the suggestion to move the "Study Area" section to the Discussion, we have chosen to retain it in the Materials and Methods section because it provides essential geographical and climatic context that directly informs the modeling framework. This approach is also consistent with previous studies using ecological niche modeling, where the description of the study area is typically presented early to support variable selection and model interpretation. We believe this placement ensures clarity for readers in understanding the spatial and environmental background before evaluating the results.

Lines 260-270 - There is mistake in description of the colours.

Response: Thank you for pointing this out. We have corrected the description to accurately match the color scheme used in the map, ensuring consistency between the text and figure legend. The updated text now reads: “This map displays a continuous probability scale, ranging from low suitability (represented in green) to high suitability (indicated in red), which reflects the likelihood of LSD presence based on current environmental conditions. The model identifies extensive regions of high suitability for LSD occurrence, particularly in the central and northeastern areas, which are depicted in orange to red shades. Moderate suitability is noted in certain regions of the southern provinces, while areas with low suitability, represented in green, are located in the northern, western, and some parts of the southern regions.” (Line 280-287)

There is a discrepancy in the statements:

Lines 224 – 226 – “The NDVI demonstrated a non-linear positive association with LSD risk, with the highest predicted probability occurring in areas with moderate levels of vegetation.”

Line 306-308 – “Areas with greater vegetation density may offer favorable conditions for vector survival and reproduction, thereby contributing to the spatial distribution of LSD risk [25, 44-45].”

What is your opinion?

Response: Thank you for highlighting this important point. We acknowledge that the statements may initially appear inconsistent, but they actually reflect two complementary observations.

The model’s response curve (Lines 244 – 245) showed that LSD suitability was highest in areas with moderate NDVI values, rather than at the extremes. This suggests that overly dense vegetation may not be optimal for vector activity, while moderate vegetation likely provides a balance of shade, humidity, and accessibility that supports vector survival.

The second statement (Lines 330 to 338) refers more generally to how areas with more vegetation, compared to very sparse or barren environments, tend to support vector populations because they offer better microclimatic conditions. To avoid confusion, we have revised the text into “Areas with moderate vegetation density may offer favorable conditions for vector survival and reproduction, thereby contributing to the spatial distribution of LSD risk.”

Again, we sincerely thank the reviewer for painstakingly evaluated of our manuscript and have provided constructive comments. 

Reviewer 3 Report

Comments and Suggestions for Authors

Refer to the attached

Author Response

Reviewer 3

Reviewer Comments on Manuscript

Response: Overall, we sincerely thank the reviewer for painstakingly evaluated of our manuscript and have provided constructive comments. We are truly grateful for the time and effort dedicated to reviewing our work and for the valuable feedback and suggestions rendered to us.

Simple Summary

Line 23 The summary notes "moderate accuracy" but does not specify what threshold defines “moderate” AUC in ecological niche modeling. Could the author briefly clarity the performance benchmark and implications for reliability benchmark and implications for reliability?

Response: Thank you for this helpful comment. To improve clarity, we have revised the sentence to explain what constitutes moderate accuracy in the context of AUC values. The updated text from “The model has AUC value of 0.699 (~0.70) indicating moderate predictive ability greater than random chance, and based on this, the model’s performance supports its reliability in identifying environmentally suitable areas for LSD. Central and northeastern region were defined as a high suitability area of LSD outbreak” (line 23-26). This clarification helps avoid confusion regarding the interpretation of model accuracy and its implications for reliability.

Abstract

Line 35 The AUC value (0.699) is reported, but the abstract does not discuss its practical meaning in predictive epidemiology. Could the authors briefly explain the interpretation in this context?

Response:

Thank you for this valuable suggestion. We have revised the simple summary and abstract to briefly explain the practical meaning of the AUC value in the context of predictive epidemiology (Please see line 23-25).

Line 37 consider specifying the relative

contributions (e.g., land cover = ###%?, cattle density = ###%?) to strengthen the abstract's informativeness.

Response: Thank you for this helpful suggestion. We have included the relative contribution such as “land cover (65%), cattle density (25%), and NDVI (3%)” (line 35-37).

Introduction

Lines 66 71 The introduction would be strengthened by integrating recent experimental findings on LSDV pathogenesis. Consider referencing the 2025 experimental study on LSDV pathogenesis in indigenous cattle (Pathogens, doi.org/10.3390/pathogens14060577), which demonstrated strain-specific virulence and host-adaptive factors. This recommendation will enrich your discussion by connecting ecological risk patterns with underlying pathogen biology.

Response: Thank you for this valuable suggestion. We agree that integrating recent experimental findings on LSDV pathogenesis would strengthen the manuscript. However, this information is not included in the current paragraph because it focuses specifically on the situation in Thailand. We believe the suggested reference is more appropriately placed in the first paragraph, which introduces a broader perspective on LSD. (see line 50-51)

Lines 81 85 The statement about model applications in Iran, Russia, and China could benefit from specific outcomes or predictive success rates from those studies to better justify methodological choice.

Response: Thank you for this helpful suggestion. We have revised the manuscript to briefly highlight the relevance of previous MaxEnt applications in Iran, Russia, and China. These studies demonstrated the model’s consistency and adaptability in capturing environmental risk patterns across different ecological contexts, supporting our choice of MaxEnt as a suitable approach for the spatial risk assessment of LSD in Thailand.

Therefore, the sentence was revised from “For instance, the maximum entropy model has been employed to assess the potential spread of LSD outbreaks in countries such as Iran, Russia, and China” into “Previous studies in Iran, Russia, and China have successfully applied MaxEnt to map the ecological suitability of LSD vectors under diverse environmental conditions. These applications demonstrate the flexibility and consistency of the method in predicting spatial risk patterns, making it a relevant and appropriate choice for assessing LSD risk across the varied landscapes of Thailand” (Line 84-89).

Materials and Methods

Lines 110 118 The dataset split (2021 2023 vs. 2021 only) is mentioned, please elaborate on why 2021 alone was chosen for comparison and how differences in outbreak patterns might influence model performance.

Response: In the revision, why 2021 alone was chosen for comparison and how differences in outbreak patterns might influence model performance were addressed (Line 122-129).

Line 146 The multicollinearity threshold (>0.8) is stated could the authors justify this cut-off based on ecological modeling best practices or prior literature?

Response:  Thank you for your thoughtful comment. We have clarified the rationale for selecting the >0.8 Pearson correlation threshold. Line 159-162 showed that this cut-off is based on the literature and is widely used in ecological niche modeling to reduce multicollinearity among predictors. It is considered an effective threshold to ensure model stability while retaining ecologically meaningful variables. Several previous MaxEnt studies have applied the same criterion to improve model reliability and interpretability (Li et al., 2023; Xie et al., 2022). A supporting citation has been added to the manuscript.

Line 186 Parameter settings for MaxEnt are listed, but justification for the chosen regularization multiplier (1) is not provided. Please explain why this value was selected over other options.

Response: Thank you for this important observation. The reason for selecting a regularization multiplier of 1 is explained in line 196, where we note that this is the default value in MaxEnt and is commonly used to prevent overfitting while maintaining appropriate model complexity. This approach is supported by previous studies (Bosso et al., 2016; Li et al., 2023), which applied the same setting in ecological niche modeling. We retained this value to ensure consistency with prior research and enhance comparability of results.

Results

Line 206 The model's AUC is reported as 0.699, can the authors comment on whether additional evaluation metrics (e.g., True Skill Statistic, omission rate) were considered to validate predictive performance?

Response: Several publications on transboundary diseases such as lumpy skin disease and foot and mouth disease primarily use AUC as the main evaluation metric, and we followed this approach. However, we agree that including additional evaluation metrics is a valuable suggestion. In the revised manuscript, True Skill Statistic and omission rate have been added to the Materials and Methods (line 205-211), Results (line 229-235), and Discussion (line 372-377) sections.

Line 226 The highest suitability is foun - please clarify potential mechanisms driving this result, as it might appear counterintuitive for a livestock disease.

Response: Thank you for this important observation. We agree that the finding of highest suitability in urban and built-up areas (MODIS Land Cover Class 13) may initially seem unexpected for a livestock disease. However, in the context of Thailand, this can be explained by the presence of many smallholder and backyard cattle farms located within or near peri-urban and urban areas. These settings often involve close human-animal interactions, relatively high cattle density, and microclimatic conditions that support vector survival, such as heat-retaining surfaces and limited implementation of vector control measures. This explanation has been addressed in the discussion section to clarify the ecological relevance of the result. (line 305-318)

Lines 249 256 The jackknife test results are presented, did the authors check for possible variable interaction effects that could influence relative contributions?

Response: We thank the reviewer for the comment. In this study, we used the jackknife test in MaxEnt to assess the individual importance of each variable; however, the method does not explicitly evaluate interaction effects between predictors. While variable correlation was checked prior to modelling to reduce multicollinearity, the relative contributions reported by MaxEnt reflect both unique and shared effects. Investigating explicit interaction effects would require alternative modelling approaches (e.g., generalized additive models or boosted regression trees) and was beyond the scope of this ecological niche modelling study.

Discussion

Lines 285 292 Land cover is identified as the most influential predictor, please elaborate on how MODIS Class 13 specifically relates to cattle vector interactions in urban/peri-urban areas.

Response: We have expanded our explanation in the discussion to clarify the relationship between MODIS Land Cover Class 13 (urban and built-up areas) and cattle and vector interactions. In Thailand, urban and peri-urban areas often contain smallholder or backyard cattle farms that are embedded within densely populated environments. These settings can create favorable conditions for disease transmission, including high cattle density in limited-space environments, limited vector control, and suitable microclimates for vector survival such as warmer temperatures and more breeding sites. This reasoning has been discussed in the manuscript (line 305-318)

Line 320 The discussion mentions temperature and humidity enhancing vector survival, consider adding quantitative thresholds (e.g., temperature/humidity ranges) from referenced studies.

Response: Thank you for this helpful suggestion. We have revised the sentence to include specific climatic thresholds that influence vector biology from reference studies. The revised sentence from “Furthermore, studies reveal that warmer temperatures and higher humidity levels can enhance vector survival and reproduction” into "Furthermore, studies reveal that temperatures between 25°C and 32°C and relative humidity levels above 60% can enhance vector survival and reproduction" (line 340-341)

Lines 342 349 The small difference between datasets is noted. Could the authors discuss whether reduced outbreak numbers in later years might affect spatial clustering patterns or bias model predictions?

Response: As mentioned in the previous response, we have discussed this point, please see line 122 to129.

References:

Bosso, L.;  Russo, D.;  Di Febbraro, M.;  Cristinzio, G.; Zoina, A., Potential distribution of Xylella fastidiosa in Italy: a maximum entropy model. Phytopathologia Mediterranea 2016, 62-72.

Kutumbetov, L., Ragatova, A., Azanbekova, M., Myrzakhmetova, B., Aldayarov, N., Zhugunissov, K., Abduraimov, Y., Nissanova, R., Sarzhigitova, A., Kemalova, N., & Issimov, A. (2025). Investigation of the Pathogenesis of Lumpy Skin Disease Virus in Indigenous Cattle in Kazakhstan. Pathogens, 14(6).

Li, S.;  Xu, L.;  Jiao, Y.;  Li, S.;  Yang, Y.;  Lan, F.;  Chen, S.;  Man, C.;  Du, L.;  Chen, Q.;  Wang, F.; Gao, H., Risk Assessment of Global Animal Melioidosis Under Current and Future Climate Scenarios. Animals (Basel) 2025, 15 (3).

Xie, C.;  Tian, E.;  Jim, C. Y.;  Liu, D.; Hu, Z., Effects of climate-change scenarios on the distribution patterns of Castanea henryi. Ecology and Evolution 2022, 12 (12), e9597.